# The Implication of Serum Autoantibodies in Prognosis of Canine Mammary Tumors

**DOI:** 10.3390/ani12182463

**Published:** 2022-09-18

**Authors:** Stephen Hsien-Chi Yuan, Shih-Chieh Chang, Pei-Yi Chou, Youngsen Yang, Hao-Ping Liu

**Affiliations:** 1Department of Veterinary Medicine, College of Veterinary Medicine, National Chung Hsing University, Taichung 40227, Taiwan; 2Division of Hematology-Oncology, Department of Internal Medicine, Taichung, Veterans General Hospital, Taichung 40705, Taiwan

**Keywords:** canine, mammary tumor, autoantibody, serum, prognosis, survival

## Abstract

**Simple Summary:**

Canine mammary tumor (CMT) is prevalent in female dogs. As tumor recurrence and metastasis occur in malignant CMT (MMT) dogs after surgery, serum prognostic biomarkers are needed to facilitate prediction of disease outcomes. We have identified CMT-associated autoantibodies, namely TYMS-AAb, IGFBP5-AAb, HAPLN1-AAb, and AGR2-AAb, which are present in serum of CMT dogs. In this study, we demonstrated that the serum AAb levels decreased at 3- and 12-months post-surgery in 11 MMT dogs. We also evaluated the correlation between the presurgical AAb level and overall survival (OS) of 90 CMT dogs after surgery. Data reveal that higher levels of IGFBP5-AAb and TYMS-AAb are significantly correlated with worse OS. On the contrary, a lower level of AGR2-AAb is correlated with poorer OS. Notably, MMT dogs presenting higher levels of TYMS-AAb and IGFBP5-AAb plus a lower level of AGR2-AAb have worst OS. This is the first study revealing an association between serum AAb levels and clinical outcomes of CMT and provides a starting point to verify the validity of utilizing this AAb panel in prediction of tumor relapse and therapy responses.

**Abstract:**

Canine mammary tumor (CMT) is the most prevalent neoplasm in female dogs. Tumor recurrence and metastasis occur in malignant CMT (MMT) dogs after surgery. Identification of serum prognostic biomarkers holds the potential to facilitate prediction of disease outcomes. We have identified CMT-associated autoantibodies against thymidylate synthetase (TYMS), insulin-like growth factor-binding protein 5 (IGFBP5), hyaluronan and proteoglycan link protein 1 (HAPLN1), and anterior gradient 2 (AGR2), i.e., TYMS-AAb, IGFBP5-AAb, HAPLN1-AAb, and AGR2-AAb, respectively, by conducting serological enzyme-linked immunosorbent assays (ELISA). Herein we assessed serum AAb levels in 11 MMT dogs before and after surgery, demonstrating that IGFBP5-AAb and HAPLN1-AAb significantly decrease at 3- and 12-months post-surgery (*p* < 0.05). We evaluated the correlation between the presurgical AAb level and overall survival (OS) of 90 CMT dogs after surgery. Kaplan-Meier survival analysis reveals that IGFBP5-AAb^HIgh^ and TYMS-AAb^High^ are significantly correlated with worse OS (*p* = 0.017 and *p* = 0.029, respectively), while AGR2-AAb^Low^ is correlated with somewhat poorer OS (*p* = 0.086). Areas under a time-dependent receiver operating characteristic curve (AUC) of IGFBP5-AAb and TYMS-AAb in predicting OS of MMT dogs are 0.611 and 0.616, respectively. Notably, MMT dogs presenting TYMS-AAb^High^/IGFBP5-AAb^High^/AGR2-AAb^Low^ have worst OS (*p* = 0.0004). This study reveals an association between the serum AAb level and CMT prognosis.

## 1. Introduction

Canine mammary tumor (CMT) is the most prevalent neoplasm in female dogs, and approximately 50% of CMTs are malignant [1]. A prospective 2-year follow-up study reveals that rates of recurrences and distant metastases of CMT after mastectomy are 18% and 47%, respectively [2]. Nowadays, prognostic evaluation of CMT mainly relies on tumor characteristics such as histological grades [2,3], lymph node involvement [4], and expression of certain tumor-associated molecules which are immunohistochemically detected in tumor or adjacent tissues [5]. However, the required methodologies are relatively invasive and time-consuming [6]. To assist in prediction of the disease outcome, identification of prognostic biomarkers which are easily accessible in the sera of CMT patients paves a tempting avenue. Proposed serum biomarkers include the dysregulated proteins which are released from tumor cells via secretion or unconventional pathways during the development and progression of tumors [7,8]. Additionally, circulating immunoregulatory factors arising from the tumor-immune interplay have been proposed for prognosticating the outcomes of CMT [9] and related pet animals’ neoplasms [10,11].

Proteins overexpressed or aberrantly localized in tumor cells likely act as autologous immunogenic antigens, so-called tumor-associated antigens (TAAs). Humoral immune responses toward TAAs result in production of tumor-associated autoantibodies (AAbs) which are less subject to proteolysis and highly stable in serum compared to soluble TAAs. As a result, tumor-associated AAbs can persist at lasting levels in serum even when corresponding TAAs are no longer detectable, and have been proposed as serum biomarkers for predicting cancer outcomes and patients’ survival in various types of human cancer [12,13]. Despite the clinical potential of AAb, the prognostic value of utilizing single AAb could be limited due to tumor heterogeneity [14]. In this regard, it has brought increased attention to identifying a tailored panel of tumor-associated AAbs for better prognostic prediction [12].

We previously identified thymidylate synthetase (TYMS), insulin-like growth factor-binding protein 5 (IGFBP5), hyaluronan, and proteoglycan link protein 1 (HAPLN1), and anterior gradient 2 (AGR2) as CMT-associated proteins overexpressed in malignant CMT (MMT) tissues compared with normal counterparts [15]. TYMS is an enzyme essential for catalyzing the synthesis of deoxythymidine monophosphate (dTMP). Overexpression of TYMS is associated with poor prognosis and treatment resistance in human cancer [16,17,18]. IGFBP5 plays a critical role in mammary gland development [19] and regulates cell proliferation, migration, and adhesion in human breast cancer [20]. Increased abundance of IGFBP5 is associated with poor prognosis in several types of human cancer [21,22]. HAPLN1 is known for its role in structural support in extracellular cartilage matrix. Overexpression of HAPLN1 is correlated with tumor progression of mesothelioma [23]. Moreover, age-related changes in HAPLN1 increase lymphatic permeability and thus influence the route of melanoma metastasis [24].

We have verified TYMS, IGFBP5, and HAPLN1 as CMT-associated antigens and the presence of corresponding autoantibodies, namely TYMS-AAb, IGFBP5-AAb, and HAPLN1-AAb, respectively, in serum of CMT patients. These AAbs show validity in distinguishing early-stage MMT from benign CMT (BMT) or healthy dogs [25]. Whether these AAbs can be used for prediction of CMT outcomes has not yet been evaluated.

AGR2 acts as a protein disulfide isomerase (PDI) and mainly resides at the endoplasmic reticulum (ER) to modulate ER homeostasis and the quality control of proteins [26,27]. On the other hand, AGR2 can be presented at the cancer cell surface or secreted outside the cell to exert pro-oncogenic functions [28]. Our recent study demonstrates that serum AGR2 titer is associated with CMT progression and has an adverse correlation with patients’ overall survival (OS) [8]. Whether AGR2 confers immunogenicity to elicit reactive autoantibody (AGR2-AAb) remains to be verified.

In the present study, we first assessed serum AGR2-AAb in CMT patients and then evaluated the correlation between serum levels of CMT-associated AAbs and CMT outcomes. Furthermore, we monitored the changes in the serum levels of tailored panels of AAbs in CMT patients before and after tumor resection by mastectomy.

## 2. Materials and Methods

### 2.1. Serum Collection

Serum samples were collected at Veterinary Medical Teaching Hospital (VMTH), National Chung Hsing University (NCHU), Taichung, Taiwan, from 2017 to 2021. All experimental procedures and sample collection were approved by Institutional Animal Care and Use Committee (IACUC) of NCHU (Code: 109-002). Written informed consent was provided by each dog owner for the experimental procedures. Blood samples were collected from 27 healthy female dogs and 90 female CMT dogs before partial or complete mastectomy, including 20 dogs with benign CMT (BMT) and 70 dogs with malignant CMT (MMT) which were classified and staged according to the modified WHO TNM classification for CMT [1]. To monitor the serum level of CMT-associated AAb before and after surgery, another 11 dogs diagnosed with MMT were enrolled and their presurgical and postsurgical sera were collected before and at 3 and/or 12 months after surgery, respectively. All blood samples were left at room temperature (RT) for 30 min for clotting, and were centrifuged at 2500× *g*, 4 °C for 15 min. The resulting supernatants (sera) were collected and supplemented with a protease inhibitor cocktail (VWP Life Science, Avantor, Radnor, PA, USA). Sera were distributed into aliquots of 50 μL and stored at −80 °C until utilization.

### 2.2. Preparation of Recombinant Canine Proteins

Recombinant canine proteins of TYMS, IGFBP5, HAPLN1, and AGR2 were prepared as previously described [8,25]. In brief, cDNA derived from CMT tissues or CMT cell lines was used as template to amplify DNA fragments encoding the full-length canine HAPLN1, canine IGFBP5 (with deletion of the signal peptide), and canine AGR2 (with deletion of the signal peptide) by polymerase chain reaction (PCR) with forward and reverse primer pairs targeting to canine *HAPLN1* (NCBI Gene ID: 488921), *IGFBP5* (NCBI Gene ID: 610316), and *AGR2* (NCBI Gene ID:482333), respectively. The resulting DNA fragment was inserted into the pET-24a(+) expression vector via the Nde I and Xho I restriction enzyme sites. The DNA fragment for the full-length canine *TYMS* gene (Gene ID: 607417) was synthesized (Protech, Taiwan) and cloned into the pET-24(+) expression vector via the Hind III and Xho I sites. The nucleotide sequences of all the constructs were confirmed by automatic DNA sequencing (Tri-I Biotech Inc., Taiwan). The expression vector for individual recombinant protein was introduced into *Escherichia coli* (*E. coli*) BL21(DE3) for expression of recombinant proteins by induction with 0.5 mM isopropyl β-d-1-thiogalactopyranoside (IPTG). The recombinant proteins were further purified by using the nickel-nitrilotriacetic acid (Ni-NTA) Sepharose^TM^ 6 Fast Flow resins (GE healthcare, Chicago, IL, USA) and dialyzed with 1× PBS at 4 °C overnight. Protein concentrations of the recombinant proteins were determined with a Bicinchoninic acid (BCA) protein assay kit (Pierce, Thermo Fisher Scientific, Waltham, MA, USA).

### 2.3. Immunoblotting for Serum AAb Detection

Recombinant proteins were resolved in 1× sampling buffer (50 mM Tris-HCl, 1% β-mercaptoethanol, 2% SDS, 10% glycerol, 0.02% bromophenol blue, 50 mM EDTA, pH 6.8), denatured at 95 °C for 10 min and subsequently separated by sodium dodecyl sulfate polyacrylamide gel electrophoresis (SDS-PAGE) with 4%-to-20% gradient polyacrylamide gels which were casted with a gradient gel solution (BIOTOOLS, Taipei, Taiwan). Proteins resolved in gels were transferred onto polyvinylidene difluoride (PVDF) membranes (GE Healthcare), which were blocked with HyBlock 1-min blocking buffer (GOAL Bio, Hycell International Co., Ltd., Taipei, Taiwan) at RT for one minute. The membranes were incubated with serum samples collected respectively from a malignant CMT dog and a healthy control at 50-fold dilution in HyBlock 1-min. Serum AAbs reactive to the recombinant proteins were sequentially detected by a goat anti-canine IgG antibody conjugated with horseradish peroxidase (HRP; PerkinElmer, Waltham, MA, USA) at 1:10,000 dilution. For positive controls, recombinant proteins were blotted with a rabbit IgG specific to AGR2 (Invitrogen, Thermo Fisher Scientific, Catalogue No. MA5-16244) at 1:500 dilution, to TYMS (Proteintech, Rosemont, IL, USA, Catalogue No. 15047-1-AP) at 1:1000 dilution, to HAPLN1 (Invitrogen, Thermo Fisher Scientific, Catalogue No. MA5-24856) at 1:1000 dilution, or to IGFBP5 (Proteintech, Catalogue No. 55205-1-AP) at 1:1000 dilution, followed by detection with a goat anti-rabbit IgG antibody conjugated with HRP (PerkinElmer) at 1:10,000 dilution. The blotted proteins were illuminated using Western Lightning ECL-Pro (PerkinElmer). The images were acquired using a Luminescence Imager System (Hansor Polymer Technology Corp., Taichung, Taiwan) equipped with the TSGel software (version 3.5).

### 2.4. Procedures of an Indirect Enzyme-Linked Immunosorbent Assay (ELISA) for Serum AAb Detection

An in-house ELISA was established for serum AAb detection [25]. The recombinant proteins diluted in 1× PBS were coated onto a 96-well microplate (ExtraGene, Davis, CA, USA) at a concentration of 250 ng/100 µL per well at 4 °C for 16 h. The wells were washed with 1× PBS (200 µL/well) five times and blocked with BlockPRO^TM^ (Visual Protein, Taipei, Taiwan; 200 μL/well) at RT for 1 h. After washing with 1× PBS five times, the wells were incubated with canine serum samples 30-fold diluted in 100 µL BlockPRO^TM^ at RT for 1 h, followed by washing with 1× PBS-T (0.05% Tween 20; VWR International, Avantor, Radnor, PA, USA) five times. The wells were incubated with 150 µL of HRP-conjugated goat anti-dog IgG (PerkinElmer, Waltham, MA, USA) diluted at 1:10,000 in BlockPRO^TM^ at RT for 40 min and then washed with 1× PBS-T five times. The reaction was developed by incubating with tetramethylbenzidine (TMB; Clinical Science Products Inc., Mansfield, MA, USA; 100 μL/well) in the dark for 5 min, and was stopped by adding 2N sulfuric acid (50 μL/well). Optical density (OD) values of the reactions were measured using SPECTRO star Nano (BMG LABTECH, Ortenberg, Germany) at the wavelength of 450 nm. The OD_450_ values in individual wells were corrected by subtracting the OD_540_ values of the same wells following the instruction on the TMB substrate. All the samples were assayed in duplicate. The data were presented as [mean (OD_450_ − OD_540_) of samples − mean (OD_450_ − OD_540_) of the blank control].

### 2.5. Follow-Up of CMT Patients

Ninety CMT dogs were periodically followed up through telephone interviews every 6 months. Overall survival time (OST) was calculated as the survival period from surgery to death or to the last of follow-up visit (alive). The patients who died within one week after surgery or died from an unrelated cause (e.g., accidence or heart diseases) were excluded from this study. No patient was performed with euthanasia because of CMT.

### 2.6. Statistical Analysis

Comparison of presurgical serum AAb levels between two categories was conducted using the Mann–Whitney *U* test. Difference of serum AAb levels before and after surgery was analyzed by paired Wilcoxon matched-pairs signed-rank test and Wilcoxon signed-rank test. For overall survival (OS) analysis, the median serum levels (OD) of TYMS-AAb (0.370), IGFBP5-AAb (0.313), HAPLN1-AAb (0.344), and AGR2-AAb (0.274) were respectively set as the cutoff to dichotomize the analyzed cases into high (the AAb level > median) and low (the AAb level ≤ median) groups. OS analysis was conducted using the Kaplan-Meier method, and the statistical significance was evaluated using the log-rank test. All statistical analyses were performed by using the GraphPad Prism V8.4 software (GraphPad Inc., San Diego, CA, USA). All *p* values were two-tailed; a *p* < 0.05 was considered statistically significant. The performance of the serum AAb level in OS prediction was estimated by using the “survival ROC” package with a time-dependent ROC curve in RStudio V1.1463 software (RStudio, Inc., Boston, MA, USA) as previously described [29].

## 3. Results

### 3.1. Association of Serum AAb Levels with Malignancy and Progression of CMT

To verify the correlation between CMT progression and serum levels of TYMS-AAb, IGFBP5-AAb, HAPLN1-AAb, and AGR2-AAb, we conducted an in-house ELISA established previously [25] to assess the AAbs in sera collected from 20 BMT, 70 MMT, and 27 healthy dogs. Signalments of 90 CMT dogs were summarized in Table 1. The median and mean of individual AAb levels were summarized in Appendix A. The antigen specificity of individual serum AAb was confirmed by immunoblotting analysis (Appendix A).

Among the assessed AAbs, TYMS-AAb and IGFBP5-AAb were at higher levels in MMT dogs compared either with those in healthy controls (*p* < 0.001 and *p* < 0.05, respectively) or with those in the BMT group (*p* < 0.05, Figure 1A,B, left panels). Higher levels of HAPLN1-AAb were also observed in MMT dogs compared with the healthy controls (*p* < 0.05, Figure 1C, left panel). However, AGR2-AAb levels were not distinguishable between any given group (Figure 1D, left panel).

In respect of CMT progression, serum TYMS-AAb and IGFBP5-AAb levels were significantly higher in dogs with non-metastatic MMT (stage I/II/III) compared with those in the BMT group (*p* < 0.05; Figure 1A,B, right panel) or with the healthy controls (*p* < 0.01 and *p* < 0.05, respectively). Moreover, HAPLN1-AAb levels in the metastasized MMT dogs were significantly higher than those in the healthy dogs (*p* < 0.01; Figure 1C, right panel). Together, these data suggest that TYMS-AAb, IGFBP5-AAb, and HAPLN1-AAb are associated with CMT malignancy, and TYMS-AAb and IGFBP5-AAb are also associated with CMT progression.

### 3.2. Declined Serum Levels of CMT-Associated AAbs after CMT Resection

To evaluate the responses of CMT-associated AAbs after tumor resection, serum samples of 11 MMT dogs were collected before surgery and at 3 and/or 12 months post-surgery, respectively. Signalments of these 11 MMT cases are listed in Appendix A. As shown in Table 2, levels of TYMS-AAb, IGFBP5-AAb, and HAPLN1-AAb all declined after tumor resection. Notably, serum IGFBP5-AAb and HAPLN1-AAb levels decreased most significantly at 3 months after surgery (IGFBP5, *p* = 0.0078; HAPLN1, *p* = 0.027) and remained significantly low compared with the pre-surgery levels (IGFBP5-AAb, *p* = 0.031; HAPLN1-AAb, *p* = 0.032).

To better monitor the changes in serum AAb levels in individual patients after surgery, a postsurgical-to-presurgical ratio of the serum AAb was calculated by dividing the postsurgical AAb levels by the matched presurgical levels. As shown in Figure 2, ratios of the three AAbs, in particular IGFBP5-AAb and HAPLN1-AAb, noticeably declined at 3 and 12 months after tumor resection (Figure 2B,C, respectively), albeit the reduction in the ratio of TYMS-AAb was not statistically significant (Figure 2A). These data reinforce the correlation between serum levels of CMT-associated AAbs and CMT growth.

### 3.3. Correlation between Presurgical Serum AAb Level and Overall Survival Time of CMT Dogs

We further investigated the prognostic correlation between the presurgical serum AAb level and CMT patient’s outcomes in terms of overall survival time (OST). The median and the range of OST of 90 CMT dogs or those of the subgroups were summarized in Table 3.

The median serum level of individual AAb was set as the cutoff to dichotomize 90 CMT dogs into two groups, i.e., TYMS-AAb^High^ (*n* = 49) and TYMS-AAb^Low^ (*n* = 41); IGFBP5-AAb^High^ (*n* = 48) and IGFBP5-AAb^Low^ (*n* = 42); AGR2-AAb^High^ (*n* = 46) and AGR2-AAb^Low^ (*n* = 44); HAPLN1-AAb^High^ (*n* = 49) and HAPLN1-AAb^Low^ (*n* = 41). Kaplan-Meier survival analysis revealed that higher serum IGFBP5-AAb and TYMS-AAb levels were significantly correlated with inferior OS (*p* = 0.017, Figure 3A and *p* = 0.029, Figure 3B, respectively). On the contrary, serum AGR2-AAb levels were positively correlated with better OS, although the correlation was not statistically significant enough (*p* = 0.086, Figure 3C). By contrast, no correlation was observed between HAPLN1-AAb levels and OS (*p* = 0.683, Figure 3D).

### 3.4. Effectiveness of Utilizing the Serum AAb Levels for OS Prediction

We next exploited a time-dependent receiver operating characteristic curve (ROC) to estimate the prognostic potential of the serum AAb level in OS prediction. As listed in Table 3, the area under ROC curve (AUC) of utilizing IGFBP5-AAb level in predicting OS of CMT dogs was 0.639, superior to that of utilizing TYMS-AAb (AUC = 0.600), HAPLN1-AAb (AUC = 0.500), or AGR2-AAb (AUC = 0.491). For predicting OS of MMT dogs, the AUC of utilizing IGFBP5-AAb and that of utilizing TYMS-AAb were 0.611 and 0.616, respectively, superior to that of utilizing HAPLN1-AAb (AUC = 0.489) or AGR2-AAb (AUC = 0.467).

Furthermore, for predicting OS of non-metastasized MMT dogs (stage I/II/III), utilizing TYMS-AAb and utilizing IGFBP5-AAb achieved best AUC (0.664 and 0.661, respectively). Notably, the AUC of utilizing HAPLN1-AAb in predicting OS of non-metastasized MMT dogs also reached 0.614. Collectively, the data indicate that TYMS-AAb and IGFBP5-AAb are potentially useful in OS prediction for MMT dogs, particularly for non-metastasized patients.

### 3.5. Correlation between Combined Panels of Serum AAb Levels and OS of CMT Dogs

Given the inverse correlations of TYMS-AAb and IGFBP5-AAb levels with OS, and the positive correlation between AGR2-AAb level and OS, we next evaluated the OS of CMT dogs exhibiting high levels of TYMS-AAb (TYMS-AAb^High^) and/or high levels of IGFBP5-AAb (IGFBP5-AAb^High^), plus a low level of AGR2-AAb (AGR2-AAb^Low^). Data revealed that CMT dogs exhibiting TYMS-AAb^High^ plus AGR2-AAb^Low^ had worse OS (*p* < 0.0001, Figure 4A) compared with the others. Likewise, CMT dogs showing IGFBP5-AAb^High^ plus AGR2-AAb^Low^, and those showing TYMS-AAb^High^ plus IGFBP5-AAb^High^ also had unfavorable outcomes (*p* = 0.0054, Figure 4B and *p* = 0.0098, Figure 4C, respectively). Furthermore, CMT dogs exhibiting TYMS-AAb^High^, IGFBP5-AAb^High^, plus AGR2-AAb^Low^ had significantly adverse OS (*p* < 0.0001, Figure 4D). Results reinforce the association of serum TYMS-AAb, IGFBP5-AAb, and AGR2-AAb levels with CMT prognosis.

### 3.6. Correlation between the Combined Panels of Serum AAb Levels and OS of MMT Dogs

We then evaluated the prognostic potential of combination of serum AAb levels in OS prediction for MMT dogs. Kaplan-Meier survival analysis demonstrated that MMT dogs presenting TYMS-AAb^High^ plus AGR2-AAb^Low^ had significant unfavorable OS (*p* = 0.007, Figure 5A). Moreover, MMT dogs presenting IGFBP5-AAb^High^ and AGR2-AAb^Low^, and those presenting IGFBP5-AAb^High^ plus TYMS-AAb^High^ also had adverse OS (*p* = 0.016, Figure 5B and *p* = 0.049, Figure 5C). Of note, MMT dogs exhibiting TYMS-AAb^High^, IGFBP5-AAb^High^, and AGR2-AAb^Low^ had worst OS (*p* = 0.0004, Figure 5D).

Based on the greater effectiveness of utilizing the serum AAb levels for predicting OS of MMT dogs in non-metastatic stages (Table 3), we subsequently evaluated the prognostic potential of combination of serum AAb levels for non-metastasized MMT dogs. Patients showing TYMS-AAb^High^ plus AGR2-AAb^Low^ had worse OS (*p* = 0.0005, Figure 6A), and those presenting IGFBP5-AAb^High^ plus AGR2-AAb^Low^ also had poorer outcomes (*p* = 0.041, Figure 6B). Finally, the patients exhibiting TYMS-AAb^High^, IGFBP5-AAb^High^, plus AGR2-AAb^Low^ had worst OS (*p* = 0.0006; Figure 6C). Data reveal that a combination of TYMS-AAb^High^ or IGFBP5-AAb^High^ with AGR2-AAb^Low^ is potentially useful for prognosing the disease outcomes of non-metastasized MMT dogs.

## 4. Discussion

Despite the high incidence of CMT recurrences and metastases, studies centered on the development of serological assays for prognosis of CMT are relatively limited in contrast to human neoplastic diseases. Tumor-associated AAbs have been proposed as prognostic biomarkers that can predict cancer recurrence, metastasis, and patient survival in various types of human cancer. Previously, we have identified that higher serum levels of TYMS-AAb, IGFBP5-AAb, and HAPLN1-AAb are associated with early-stage MMT and are able to discriminate between BMT and MMT [25]. In the present study, we extended the investigation to confirm that serum levels of TYMS-AAb, IGFBP5-AAb, and HAPLN1-AAb are elevated in MMT dogs compared to healthy dogs. Furthermore, we monitored the changes in serum levels of the CMT-associated AAbs in MMT dogs before and after tumor resection. Compared with the presurgical levels, serum levels of TYMS-AAb, IGFBP5-AAb, and HAPLN1-AAb decrease at 3 months after surgery and remain low up to one year, implying a possibility of utilizing CMT-associated AAbs for predicting tumor recurrence or treatment responses. Along the same lines, several studies have reported that levels of tumor-associated AAbs decrease after surgically tumor removal in human breast cancer [30,31]. Further investigations using a large-scale cohort of samples are needed to validate the clinical value of CMT-associated AAbs in tumor recurrence prediction.

In the present study, we followed up OS of CMT dogs after surgical removal of tumor and explored the correlation between the presurgical serum level of CMT-associated AAb and OS. Our data reveal an association between higher serum levels of TYMS-AAb and IGFBP5-AAb and worse OS. Overexpression of TYMS and IGFBP5 has been reported to be associated with shorter survival time and poorer disease outcomes in human pancreatic cancer [16] and urothelial carcinoma [20], respectively. It is likely that higher levels of TYMS-AAb and IGFBP5-AAb in CMT dogs are a result of expansion of humoral immune responses toward TYMS and IGFBP5 which are overexpressed in CMT tissues and deteriorate the disease outcomes. Likewise, it has been reported that tumor-associated AAbs against GRP78 (glucose-regulated protein 78) actively promote tumorigenesis and tumor progression [32]. On the contrary, serum AGR2-AAb level is higher in healthy dogs and positively associated with favorable OS of CMT dogs. Similarly, Tabuchi and colleagues demonstrate that healthy controls have a significantly higher titer of autoantibodies against human epidermal growth factor receptor 2 (HER2) compared to patients with invasive breast carcinoma [33]. The authors also suggest that HER2-AAb-positive patients have a significantly better recurrence-free survival (RFS) compared with the HER2-AAb-negative group. These findings reflect varied effects of humoral responses toward TAAs. With regard to cancer prognostic application, there is a need to differentiate the protective and pathogenic roles of tumor-associated AAbs.

Due to tumor heterogeneity and variable profiles of tumor-associated AAbs among patients, it could be less effective to use a single AAb for cancer prognostication [14,34]. Detection of multiple tumor-associated AAbs simultaneously may provide an opportunity to overcome the biological heterogeneity of tumor and tumor microenvironment. Our data reveal that CMT dogs presenting TYMS-AAb^High^/AGR2-AAb^Low^, those presenting IGFBP5-AAb^High^/AGR2-AAb^Low^, and those presenting TYMS-AAb^High^/IGFBP5-AAb^High^ have worse OS (*p* < 0.0001, *p* = 0.0054, and *p* = 0.0098, respectively) when compared with the others. Notably, CMT dogs exhibiting TYMS-AAb^High^, IGFBP5-AAb^High^ plus AGR2-AAb^Low^ have worst OS (*p* < 0.0001). Moreover, a combination of TYMS-AAb^High^, IGFBP5-AAb^High^ and AGR2-AAb^Low^ also stratifies MMT dogs having poorest OS (*p* = 0.0004). These data reveal a correlation of serum levels of TYMS-AAb, IGFBP5-AAb, and AGR2-AAb with CMT outcomes, suggesting a possibility of utilizing this AAb panel for prognostic prediction of CMT. As there is a limited number of prognostic AAb identified for CMT, it is critically needed to expand the panel of CMT-associated AAbs possessing prognostic potential.

## 5. Conclusions

In summary, the present study is the first to demonstrate an association of circulating TYMS-AAb, IGFBP5-AAb, and AGR2-AAb with CMT prognosis. This study provides a starting point to further verify the validity of this tailored AAb panel in prediction of tumor relapse and therapy responses of CMT in the future.

## Figures and Tables

**Figure 1 animals-12-02463-f001:**
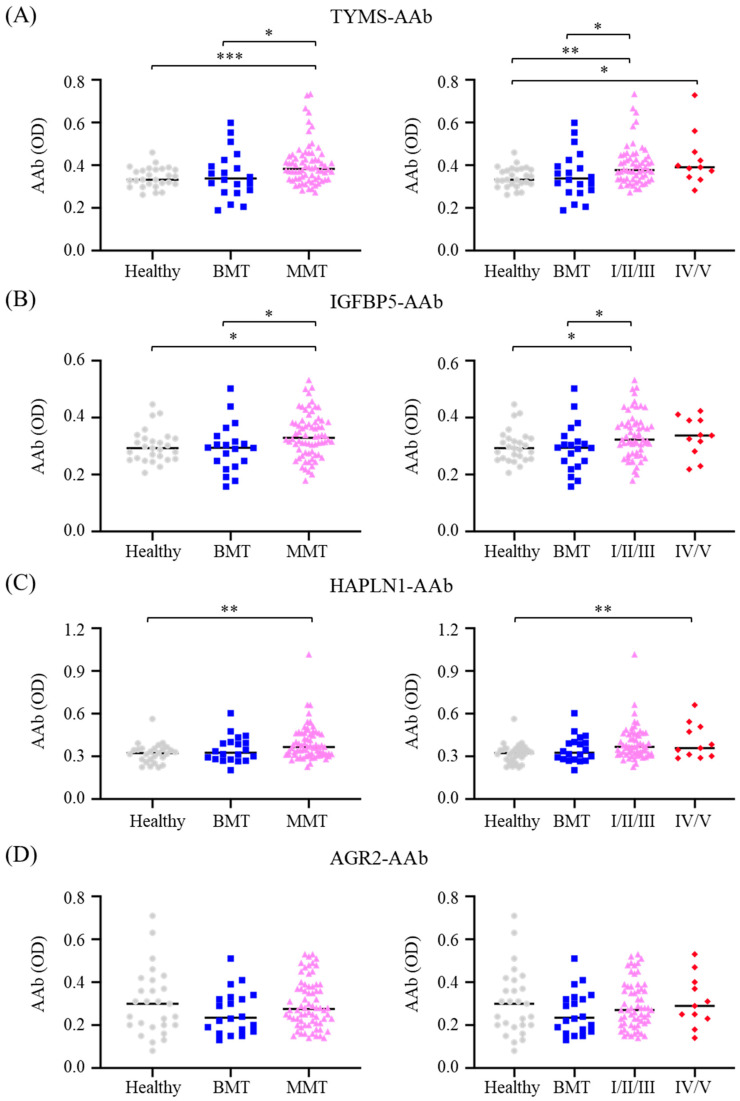
Comparison of serum levels of CMT-associated autoantibodies (AAbs) between CMT dogs and healthy controls. Serum levels of (**A**) TYMS-AAb, (**B**) IGFBP5-AAb, (**C**) HAPLN1-AAb, and (**D**) AGR2-AAb in dogs with malignant CMT (denoted MMT; left panels), which comprised the non-metastatic group (stage I, II, and III; denoted I/II/III; right panels) and the metastatic group (stage IV and V; denoted IV/V; right panels), were compared with those in the group of benign CMT (denoted BMT) or healthy controls. Results are presented in scatter dot plots, wherein each dot represents the serum AAb level in individual patients. The bar shown is the median. Statistical analysis was conducted with the Mann–Whitney *U* test. * *p* < 0.05; ** *p* < 0.01; *** *p* < 0.001.

**Figure 2 animals-12-02463-f002:**
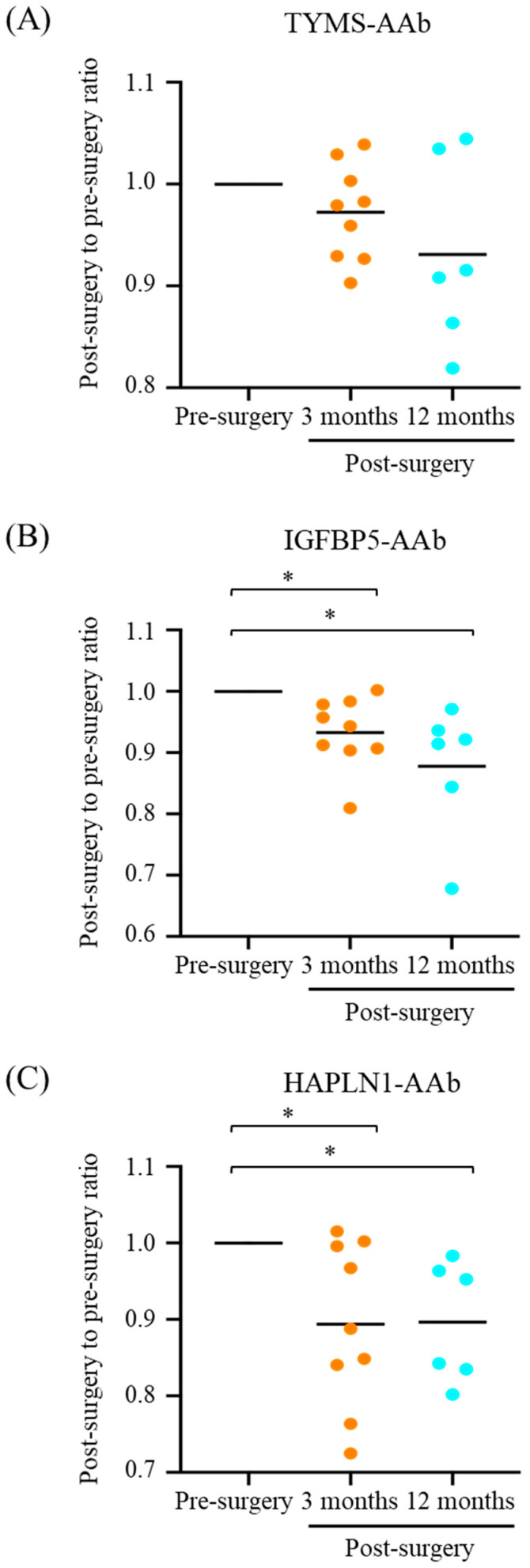
Serum levels of CMT-associated AAbs in dogs with malignant CMT (MMT) before and after mastectomy. Serum levels of (**A**) TYMS-AAb, (**B**) IGFBP5-AAb, and (**C**) HAPLN1-AAb in MMT dogs assessed at 3 and/or 12 months after surgery were compared with the corresponding serum AAb levels before surgery. A postsurgical-to-presurgical ratio of individual serum AAb was determined by dividing the postsurgical levels of individual serum AAbs by the matched presurgical levels for each patient. Results are presented in scatter dot plots, where the bar indicates the median. Statistical analysis was conducted with Wilcoxon signed-rank test. * *p* < 0.05.

**Figure 3 animals-12-02463-f003:**
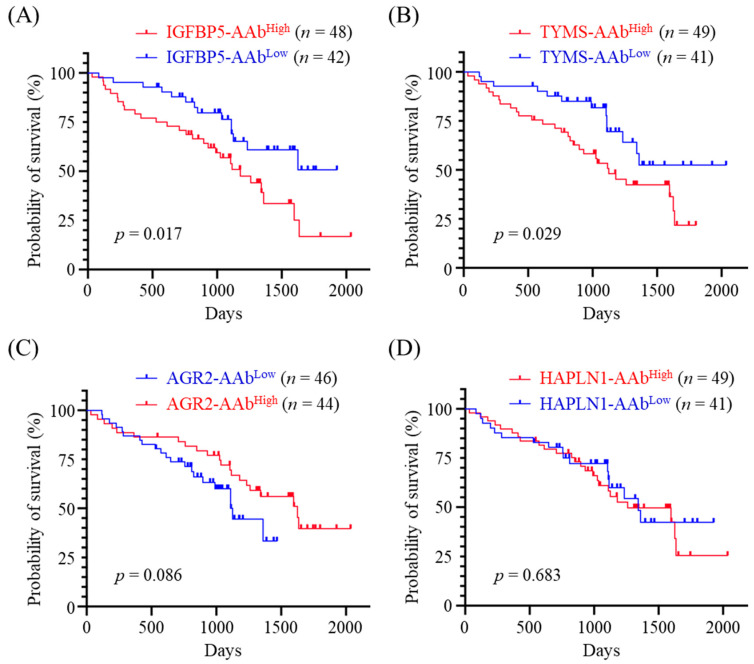
Correlation between serum AAb levels and overall survival time (OST) of CMT dogs. Ninety CMT dogs were divided into two groups exhibiting low and high serum levels of (**A**) IGFBP5-AAb, (**B**) TYMS-AAb, (**C**) AGR2-AAb, and (**D**) HAPLN1-AAb, respectively. Comparison of OST between the two groups was carried out by using Kaplan-Meier analysis. Statistical significance was determined by the log-ranked test.

**Figure 4 animals-12-02463-f004:**
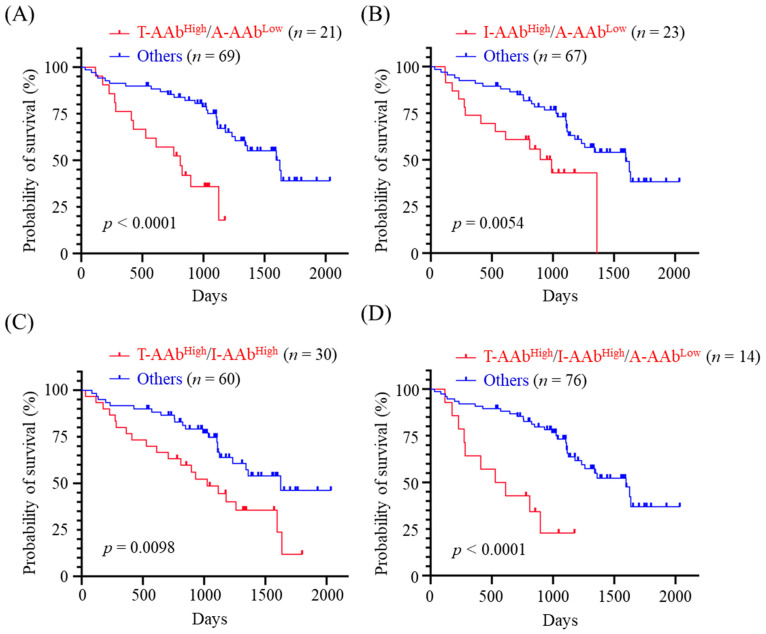
Correlation between the combined panels of serum AAb levels and OST of CMT dogs. CMT dogs (*n* = 90) were dichotomized by a combination of (**A**) TYMS-AAb^High^ and AGR2-AAb^Low^ (denoted T-AAb^High^/A-AAb^Low^), (**B**) IGFBP5-AAb^High^ and AGR2-AAb^Low^ (denoted I-AAb^High^/A-AAb^Low^), (**C**) TYMS-AAb^High^ and IGFBP5-AAb^High^ groups (denoted T-AAb^High^/I-AAb^Low^), and (**D**) TYMS-AAb^High^, IGFBP5-AAb^High^, and AGR2-AAb^Low^ (denoted T-AAb^High^/I-AAb^High^/A-AAb^Low^), respectively. OST of CMT dogs was compared between the dichotomous groups by using Kaplan-Meier analysis. Statistical significance was determined by the log-ranked test.

**Figure 5 animals-12-02463-f005:**
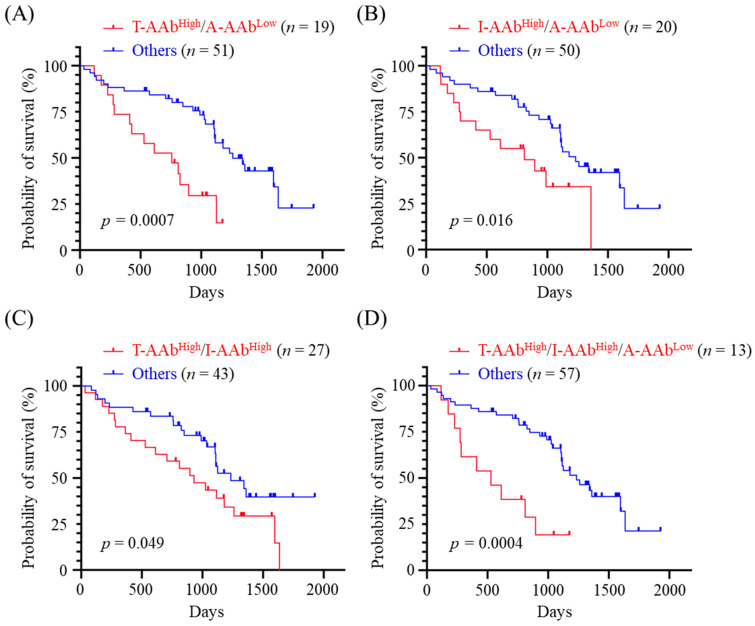
Correlation between the combined panels of serum AAb levels and OS of MMT dogs. Dogs with MMT (*n* = 70) were dichotomized by a combination of (**A**) TYMS-AAb^High^ and AGR2-AAb^Low^ (denoted T-AAb^High^/A-AAb^Low^), (**B**) IGFBP5-AAb^High^ and AGR2-AAb^Low^ (denoted I-AAb^High^/A-AAb^Low^), (**C**) TYMS-AAb^High^ and IGFBP5-AAb^High^ (denoted T-AAb^High^/I-AAb^Low^), and (**D**) TYMS-AAb^High^, IGFBP5-AAb^High^, and AGR2-AAb^Low^ (denoted T-AAb^High^/I-AAb^High^/A-AAb^Low^), respectively. OST of MMT dogs was compared between the dichotomous groups by using Kaplan-Meier analysis. Statistical significance was determined by the log-ranked test.

**Figure 6 animals-12-02463-f006:**
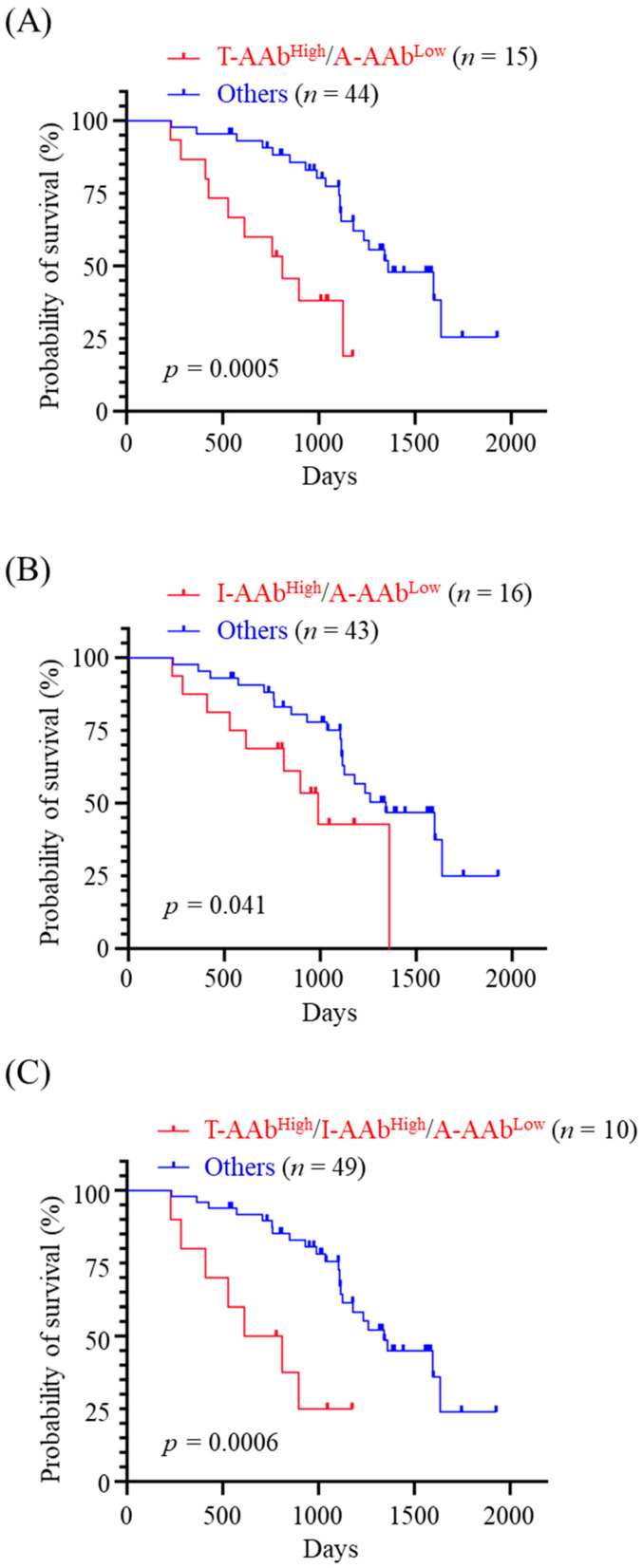
Correlation between the combined panels of serum AAb levels and OS of dogs with non-metastasized MMT. Dogs with non-metastatic MMT (*n* = 59 in stage I, II, or III) were dichotomized by a combination of (**A**) TYMS-AAb^High^ and AGR2-AAb^Low^ (denoted T-AAb^High^/A-AAb^Low^), (**B**) IGFBP5-AAb^High^ and AGR2-AAb^Low^ (denoted I-AAb^High^/A-AAb^Low^), and (**C**) TYMS-AAb^High^, IGFBP5-AAb^High^, and AGR2-AAb^Low^ (denoted T-AAb^High^/I-AAb^High^/A-AAb^Low^), respectively. OST of non-metastasized MMT dogs was compared between the dichotomous groups by using Kaplan-Meier analysis. Statistical significance was determined by the log-ranked test.

**Table 1 animals-12-02463-t001:** Clinicopathological characteristics of 90 female dogs with mammary tumors enrolled in the overall survival analysis.

CMT	Benign Mammary Tumor (BMT)			Malignant Mammary Tumor (MMT)		
Group	Case Number (%)	Median (Range) Age, Year	Median (Range) B.W., kg	Case Number (%)	Median (Range) Age, Year	Median (Range) B.W., kg
Total	20 (100)	8.4 (3–16)	5.4 (2.45–38)	70 (100)	10 (1–16)	7.4 (1.7–44.6)
Breed						
Pedigree	18 (90)	8.2 (3–15)	5.4 (2.45–38)	55 (78.6)	9 (1–16)	5.9 (1.7–38)
Mixed	2 (10)	14.5 (13–16)	18.4 (13.4–23.4)	15 (21.4)	12 (8–16)	15.3 (2.4–44.6)
Neuter status						
Yes	6 (30)	10 (9–15)	7.2 (2.45–19.4)	29 (41.4)	12 (3.5–16)	11.4 (2.1–44.6)
No	14 (70)	7.9 (3–16)	5.5 (2.7–38)	41 (58.6)	8.5 (1–13)	5.6 (1.7–30.4)
Clinical stage ^1^						
I	-	-		34 (48.6)	9 (1–16)	5.05 (1.7–20.6)
II	-	-	-	9 (12.9)	11 (8–16)	8.5 (2.1–44.6)
III	-	-		16 (22.8)	12 (8–16)	11.8 (2.36–29.8)
IV				7 (10)	13 (8–15)	10.8 (4.2–38)
V	-	-		4 (5.7)	11 (3.5–11)	19.7 (12.9–26.4)
Metastasis	-	-				
No (I/II/III)	-	-		59 (84.3)	10 (1–16)	6.9 (1.7–44.6)
Yes (IV/V)	-	-		11 (15.7)	11 (3.5–15)	15.6 (4.2–38)

^1^ Clinical stage was determined according to the modified WHO classification for domestic animals.

**Table 2 animals-12-02463-t002:** Serum levels of CMT-associated AAbs in 11 CMT dogs before and after surgery.

	Pre-Surgery	Post-Surgery			
		3 Months	*p* Value ^1,2^	12 Months	*p* Value ^1,2^
Case number	11	9	-	6	-
Serum AAb levels (mean ± SD)					
TYMS-AAb	0.534 ± 0.109	0.487 ± 0.087	0.203	0.499 ± 0.100	0.156
IGFBP5-AAb	0.281 ± 0.057	0.245 ± 0.033	0.008 **	0.253 ± 0.038	0.031 *
HAPLN1-AAb	0.248 ± 0.043	0.225 ± 0.057	0.027 *	0.230 ± 0.037	0.031 *

^1^ Comparison between postsurgical levels and presurgical levels of AAbs was determined by the paired Wilcoxon matched-pairs signed rank test. ^2^ The *p* value < 0.05 is consider statistically significant; * *p* < 0.05; ** *p* < 0.01.

**Table 3 animals-12-02463-t003:** Effectiveness of utilizing serum AAb in predicting overall survival (OS) of CMT dogs.

	CMT (BMT, MMT)	MMT (Stage I–V)	Non-Metastasized CMT (Benign, I/II/III)	Non-Metastasized MMT (I/II/III)
OST (days)				
Median	1037.5	1014.5	1103	1047
Range	33–2035	22–1927	229–2035	229–1927
AUC for OS prediction				
TYMS-AAb	0.600	0.611	0.623	0.664
IGFBP5-AAb	0.639	0.616	0.671	0.661
HAPLN1-AAb	0.500	0.489	0.591	0.614
AGR2-AAb	0.491	0.467	0.491	0.458

## Data Availability

The datasets used and/or analyzed during the current study are available from the corresponding author on reasonable request.

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
