# Peer review of "The Implication of Serum Autoantibodies in Prognosis of Canine Mammary Tumors"

_animals, 2022, doi:10.3390/ani12182463_

Round 1
Reviewer 1 Report
please important citation you lost :
Lymphatic Drainage Mapping with Indirect Lymphography for Canine Mammary Tumors
presence of regional lymph node metastases is a relevant factor affecting prognosis and treatment in cases of mammary gland tumors. The sentinel lymph node (SLN) is the first lymph node (or nodes) in the regional lymphatic basin that receives lymphatic flow from the primary neoplasm.
Author Response
#Reviewer 1:
please important citation you lost:
Lymphatic Drainage Mapping with Indirect Lymphography for Canine Mammary Tumors
presence of regional lymph node metastases is a relevant factor affecting prognosis and treatment in cases of mammary gland tumors. The sentinel lymph node (SLN) is the first lymph node (or nodes) in the regional lymphatic basin that receives lymphatic flow from the primary neoplasm.
Author's response: We have added the reference in the revised “Introduction” section (ref. 4, Line 50).
Reviewer 2 Report
Your paper demonstrated an association of the serum levels of TYMS-AAb, IGFBP5-AAb, and AGR2-AAb with CMT prognosis, proving potential value of tailored AAb panel in prediction of tumor relapse and therapy responses of CMT, but some details need to be addressed.
1. What the left panel and the right panel in Figure 1 want to express? And what’s the difference? It is recommended to add a comparison between stage I/II/III and stage IV/V.
2. Line 213-214 said HAPLN1-AAb levels were most distinguishable between 213 metastasized MMT dogs and healthy dogs. However, the P value can only indicate the credibility of the difference, but not the size of the difference.
3. Whether a comparison between the healthy group and the post-surgery group can be added in Table 2 to further illustrate the prognostic value of AAb?
4. Does the CMT in Table 3 refer to all dogs with CMT or dogs with BMT?
5. How are high and low doses defined in Figure 3-6?
Author Response
- What the left panel and the right panel in Figure 1 want to express? And what’s the difference? It is recommended to add a comparison between stage I/II/III and stage IV/V
Author's response: In the left panels of Figure 1, the CMT dogs are stratified by the malignancy of CMT and presented as the benign mammary tumor (BMT) group and the malignant mammary tumor (MMT) group. In the right panels, dogs in the MMT group are further stratified by tumor metastases, presented as the non-metastasized MMT (stages I/II/III) and the metastasized MMT (stages IV/V). To make the point clearer, we have modified the corresponding figure legend (Line 240-241). We did compare the AAb levels between the I/II/III and the IV/V groups, yet we only highlighted the between-group difference which is statistically significant.
- Line 213-214 said HAPLN1-AAb levels were most distinguishable between 213 metastasized MMT dogs and healthy dogs. However, the P value can only indicate the credibility of the difference, but not the size of the difference
Author's response: We have modified the description in the revised text (Line 230-232).
- Whether a comparison between the healthy group and the post-surgery group can be added in Table 2 to further illustrate the prognostic value of AAb?
Author's response: We are grateful for the suggestion. We would like to include the age-matched healthy controls in an expanded investigation to compare the serum AAb levels between the post-surgery group and the healthy group.
- Does the CMT in Table 3 refer to all dogs with CMT or dogs with BMT
Author's response: The CMT refers to all dogs with CMT, which comprises BMT and MMT, as used throughout the text. We have modified the descriptions in the revised Table 3 to better address the group terms.
- How are high and low doses defined in Figure 3-6
Author's response: The median serum AAb level (OD) in all analyzed dogs is set as the cutoff to dichotomize the dogs into the high AAb level and the low AAb level groups. The description is seen in the “2.6. Statistical analysis” section (Line 198-199) and in the “3.3. Correlation…..” section (Line 295) in the original version as well as in the revised version of our manuscript.
Reviewer 3 Report
The manuscript submitted by Yuan entitled "Implication of serum autoantibodies in prognosis of canine mammary tumor" aims to analyze the importance of serum levels of a set of autoantibodies in dogs with mammary tumors, before and after surgery. Despite the very high incidence of metastatic CMT disease and also recurrence, few number of studies aimed to develop serological assays to improve prognosis of dogs with mammary tumors, in a clear contrast to human oncology. Thus, this study is very original, and in the opinion of this reviewer, very relevant to the field. The different sections of the manuscript are very well-written, and the results showed, for the first time, a stistical association between serum levels of TYMS-AAb, IGFBP5-AAb and AGRA2-AAb with prognosis, raising the possibility of using a painel of antibodies to predict tumor relapse and therapy response in the near future, in dogs with MT.
However, several points should be improved before the acceptance of this manuscript for publication, as the follows:
- in the introduction section, authors should provide some data about serum immunocheck point markers recently described in queen with MC (e.g. PD-1, PD-L1, CTLA-4, VISTA) and compare the information with the present knowledge in dog.
- In the suplemmentary figure, authors should add the molecular marker at the left side of the different images.
- Did the authors check for statistical significance in dogs stratified by tumor histotype and different clinicopathological features (e.g. presence of necrotic tissue; tumor size; lymphocit infiltration); it be nice, if authors add these information.
Author Response
- in the introduction section, authors should provide some data about serum immunocheck point markers recently described in queen with MC (e.g. PD-1, PD-L1, CTLA-4, VISTA) and compare the information with the present knowledge in dog.
Author's response: We have added sentences (lines 50-54) and cited new references (ref. 7, 9, 10) in the “Introduction” section to supplement the information yet not to distract the main theme from the serum AAb.
- In the suplemmentary figure, authors should add the molecular marker at the left side of the different images.
Author's response: In the revised Figure S1, we have replaced the blot images with the whole-membrane versions with a ladder of molecular markers on the left sides. The uncropped images are provided in Figure S2.
- Did the authors check for statistical significance in dogs stratified by tumor histotype and different clinicopathological features (e.g. presence of necrotic tissue; tumor size; lymphocit infiltration); it be nice, if authors add these information.
Author's response: We are grateful for the suggestion. Unfortunately, we did not gather the mentioned information during the assessment of the serum AAb levels. We would like to check those clinicopathological features to evaluate their relationships with the serum AAb levels in the future.
Round 2
Reviewer 2 Report
The supplementary content of the article answered my question. There are only a few graphic details to note. In Figure 5, there are some black silk wire frames of unknown meaning.
Author Response
Author’s response: We have thoroughly checked the original file of Figure 5 as well as its embedded version in our manuscript, and yet there are no such wire frames noted. We speculate that the graphic frame may arise from an issue with the program or system compatibility when reviewing the manuscript.
Reviewer 3 Report
The manuscript resubmitted by Yuan et al. entitled "Implication of serum autoantibodies in prognosis of canine mammary tumor" is very improved for the first submission. In the opinion of this reviewer the article reaches a suitable form to be published in the SI "Prognostic Factors and Immune Response in Small Animal Oncology and Surgery" of Animals MDPI-Journal.
However, in the Introduction section, the author should add an additional reference (line 58) to reinforce the role of PD-1 and PD-L1 also in feline mammary carcinoma (10.3390/cancers12061386). This is a manuscript with a high number of citations and very relevant in the field.
Author Response
Author’s response: We have added the reference (ref. 10) in the “Introduction” section (line 58). The following references are shifted in order of citation as a result.